# CDKs in Sarcoma: Mediators of Disease and Emerging Therapeutic Targets

**DOI:** 10.3390/ijms21083018

**Published:** 2020-04-24

**Authors:** Jordan L Kohlmeyer, David J Gordon, Munir R Tanas, Varun Monga, Rebecca D Dodd, Dawn E Quelle

**Affiliations:** 1Molecular Medicine Graduate Program, Carver College of Medicine, University of Iowa, Iowa City, IA 52242, USA; jordan-kohlmeyer@uiowa.edu; 2The Department of Neuroscience and Pharmacology, Carver College of Medicine, University of Iowa, 2-570 Bowen Science Bldg., Iowa City, IA 52242, USA; 3The Department of Pediatrics, Carver College of Medicine, University of Iowa, Iowa City, IA 52242, USA; david-j-gordon@uiowa.edu; 4The Department of Pathology, Carver College of Medicine, University of Iowa, Iowa City, IA 52242, USA; munir-tanas@uiowa.edu; 5The Department of Internal Medicine, Carver College of Medicine, University of Iowa, Iowa City, IA 52242, USA; varun-monga@uiowa.edu (V.M.); rebecca-dodd@uiowa.edu (R.D.D.)

**Keywords:** cyclin-dependent kinase, sarcoma, cell cycle, therapeutics, retinoblastoma protein, CDK inhibitors

## Abstract

Sarcomas represent one of the most challenging tumor types to treat due to their diverse nature and our incomplete understanding of their underlying biology. Recent work suggests cyclin-dependent kinase (CDK) pathway activation is a powerful driver of sarcomagenesis. CDK proteins participate in numerous cellular processes required for normal cell function, but their dysregulation is a hallmark of many pathologies including cancer. The contributions and significance of aberrant CDK activity to sarcoma development, however, is only partly understood. Here, we describe what is known about CDK-related alterations in the most common subtypes of sarcoma and highlight areas that warrant further investigation. As disruptions in CDK pathways appear in most, if not all, subtypes of sarcoma, we discuss the history and value of pharmacologically targeting CDKs to combat these tumors. The goals of this review are to (1) assess the prevalence and importance of CDK pathway alterations in sarcomas, (2) highlight the gap in knowledge for certain CDKs in these tumors, and (3) provide insight into studies focused on CDK inhibition for sarcoma treatment. Overall, growing evidence demonstrates a crucial role for activated CDKs in sarcoma development and as important targets for sarcoma therapy.

## 1. Introduction

Sarcomas are rare, highly diverse malignancies. They account for just 1% of all adult human cancers, although their frequency is significantly greater (roughly 20%) among pediatric tumors. These lesions arise from mesenchymal tissue, where approximately 80% occur in soft tissue and 20% in bone [1]. Currently, there are over 70 subtypes that classify lesions based on tissue resemblance and molecular characteristics [2]. Two broad groups of sarcomas exist—those with simple karyotypes, often characterized by a specific, disease-driving alterations and those with complex karyotypes, where there are multiple genomic losses, gains, and amplifications [1]. Standard treatment for localized disease remains surgical resection with adjuvant radiation and/or chemotherapy used in certain types of sarcoma. Regrettably, many patients experience recurrence and metastasis, requiring systemic therapies that are unfortunately not very effective. Additionally, since these lesions are heterogeneous, responses to generalized treatments are variable and typically do not translate between different subtypes [3]. To combat sarcomas more effectively, the key pathways promoting their development and progression need to be elucidated. Recent advances suggest that activating alterations in cyclin-dependent kinase (CDK) pathways are major drivers of sarcomagenesis.

CDKs are serine/threonine kinases involved in key cellular processes, primarily cell cycle progression and transcription. As monomeric proteins, CDKs lack enzymatic activity due to a structural conformation that buries the catalytic and substrate binding domains [4]. To become active, CDKs require association with a regulatory subunit known as a cyclin, hence their designation as cyclin-dependent kinases. Humans have 20 CDKs that are classically divided into two main groups—cell cycle (CDKs 1, 2, 3, 4, 6, and 7) and transcriptional (tCDKs 7, 8, 9, 12, 13, and 19), with CDK7 contributing to both processes. Many CDKs that control cell cycle progression can bind multiple cyclins, allowing for dynamic regulation throughout the cell cycle as well as increased substrate possibilities. CDKs associated with transcription bind a single, specific cyclin, whose expression is not regulated in a cell cycle-dependent manner [5]. “Other” CDKs (5, 10, 11, 14–18, and 20) do not fit into the two canonical roles and, instead, exhibit diverse functions that are often tissue specific. For example, CDK11 variants have multiple functions in mediating transcription, mitosis, hormone receptor signaling, autophagy, and apoptosis [5,6]. Likewise, in the nervous system CDK5 promotes neurite outgrowth and synaptogenesis while in pancreatic β cells it reduces insulin secretion [7,8,9]. As CDKs control crucial processes required for cell survival and propagation, their hyperactivation (typically through mutation, gene amplification, or altered expression of their regulators) is commonly observed in cancer.

The rarity and diversity of sarcomas has slowed efforts to identify key mutations driving these cancers. In addition, sarcomas are sometimes simplistically viewed as a single entity or described in broad, unspecified terms. As our knowledge of sarcoma biology has increased, there is a growing appreciation for CDK pathway dysregulation in promoting disease progression. This review discusses current knowledge about CDK and CDK-related aberrations in the most common subtypes of sarcoma in both adult and pediatric patients. Additional consideration is given to CDK-targeted therapy in the pre-clinical setting as well as recent clinical trials.

## 2. Cell Cycle CDKs

Most of what is known about the CDKs in sarcoma pertains to those involved in cell cycle regulation, warranting a brief discussion of their roles and that of other pathway components in normal versus tumor cell proliferation. The somatic cell cycle is comprised of four phases: G_1_, the first gap period where cells prepare for DNA replication and decide whether to continue cycling or enter quiescence (G_0_, a reversible withdrawal from the cell cycle); S, where DNA is synthesized; G_2_, the second gap period where cells prepare for mitosis; and M, where cells undergo mitosis, involving both nuclear division and cytokinesis [10]. The transitions from one phase of the cell cycle to another are coordinated by CDKs and their cyclin partners. Together, these enzymatic complexes phosphorylate specific substrates to promote the signaling events needed for cell cycle progression.

During early G_1_ phase, mitogens induce expression of the D cyclins (D1, D2, and/or D3) which subsequently associate with the cyclin-D dependent kinases, CDK4 and/or its homolog CDK6. These complexes are inactive until phosphorylated at a conserved threonine residue in the T-loop region of the CDK (T172 in CDK4, T177 in CDK6) by a CDK-activating kinase (CAK), widely considered to be cyclin H-CDK7 but possibly including other kinases [11,12,13,14]. Once stimulated, cyclin D-CDK4/6 kinases phosphorylate several sites within the retinoblastoma (RB1) tumor suppressor protein (or its homologs, p107 and p130) [15]. In its hypo-phosphorylated state, RB1 actively suppresses G_1_-S progression by sequestering E2F transcription factors, which transcribe genes needed for DNA replication [16,17]. Phosphorylation of RB1 at multiple S/T residues, first by cyclin D-dependent kinases and subsequently by cyclin E-CDK2 in later G_1_ phase, completely inactivates RB1, enabling the release of E2F from RB1 complexes and activation of E2F-mediated transcription ([17], see Figure 1A). While RB1 (and its homologs, p107 and p130) is the primary target of D-CDK4/6 kinases, cyclin E-CDK2 acts on a wide variety of additional substrates to promote G_1_ and S phase. As one example, activated E-CDK2 kinase phosphorylates nucleophosmin (NPM/B23) to ensure proper centrosome duplication. Continued cellular progression through S, G_2_, and M phases is then driven by sequential activation of cyclin A-CDK2 and cyclin B-CDK1 kinases [18].

Timely inactivation of growth-promoting cyclin-CDK complexes is also critical to ensure proper cell cycling. This involves many levels of negative regulation, often triggered by phosphorylation and dephosphorylation of specific residues in cyclins and CDKs that changes their subcellular localization, protein interactions and/or ubiquitination, and proteasomal degradation. As such, the kinases, phosphatases, binding proteins, and ubiquitin ligases that control the cell cycle CDKs are frequently dysregulated in cancer (Figure 1B–C). For example, Cdc25A/B/C dual specificity phosphatases remove inhibitory S/T and Y phosphorylation on the CDKs and can be overexpressed in tumors [19,20]. More commonly, tumor cells downregulate endogenous CDK inhibitors, which fall into two classes of small molecular weight proteins. The first group of CDK inhibitors that was discovered constitutes the INK4 family (p15, p16, p18, and p19), so named because they specifically inhibit CDK4 (and CDK6). Among the INK4 proteins, loss of p16^INK4a^ (encoded by the *CDKN2A* gene, also called *INK4a*) occurs most frequently in human cancers. The second family of CDK inhibitors (p21, p27, and p57), called CIP/KIP for “CDK or kinase inhibitory proteins,” have more universal activity that antagonizes cyclin D-, E-, A-, and B-CDK complexes [21].

Loss of CDK inhibitors is a predominant event during sarcomagenesis. One of the most commonly altered CDK inhibitor genes in sarcomas is *CDKN2A*, which plays a complex and uniquely important role in tumor suppression. It is also referred to as the *INK4A/ARF* locus because it encodes not one but two genes, namely *INK4a* and *ARF* [22,23]. Located on the short arm of chromosome (chr) 9p21, this small locus produces two transcripts, *INK4a* (which encodes p16^INK4a^) and *ARF* (which encodes the alternative reading frame protein, called ARF) [24,25]. The two transcripts are regulated by distinct promoters upstream of unique first exons, exon 1α for *INK4a* and exon 1β for *ARF*, that are spliced onto common exons 2 and 3. Although they share DNA sequences, the proteins are translated in overlapping reading frames; consequently, p16^INK4a^ and ARF (p14 in humans, p19 in mice) bear no amino acid sequence identity and have unrelated functions. While p16^INK4a^ is a specific inhibitor of D-CDK4/6 kinases that promotes G_1_ phase arrest by RB1 activation, ARF has multiple targets through which it elicits cell cycle arrest, apoptosis, autophagy, and/or other tumor suppressive biological changes. The primary effector of ARF is p53, the “guardian of the genome” and the most frequently inactivated tumor suppressor gene in human cancers [26]. ARF induces p53 in response to oncogenic stress by binding to and inhibiting MDM2, an E3 ubiquitin ligase that normally restricts p53 expression [27,28]. The CDK inhibitor, p21, is a key transcriptional target of p53, providing a functional link between ARF-p53 signaling and RB1 activation [29,30]. Notably, ARF inhibits cancer through many other pathways because of its association with more than 40 cellular proteins. While some of those factors influence p53 activation, the majority contribute to p53-independent processes that significantly impact tumorigenesis [22,30,31,32,33,34].

In addition to *INK4a* and *ARF*, the 9p21 locus also contains the nearby *CDKN2B* (*INK4b*) gene, which encodes another INK4 protein, p15^INK4b^ [35]. While loss of p16^INK4a^ was initially believed to be the main target during cancer progression, the discovery of p14^ARF^ and prevalence of large deletions or silencing of the locus spanning *INK4a*, *ARF* and *CDKN2B* have highlighted the importance of these other tumor suppressors. Despite this knowledge, tumor analyses of 9p21 often fail to adequately discriminate between alterations in *INK4a* versus *ARF* at the *CDKN2A* locus and frequently underestimate potential contributions of *INK4b*. This is especially true for less studied, rare tumors like sarcomas where many subtypes display 9p21 deletions but the relative importance of each gene to tumor pathogenesis remains poorly understood. This is beginning to change for some of the better studied sarcomas, such as malignant peripheral nerve sheath tumors (MPNSTs), although more work is needed. The development of mouse models with selective, tissue specific inactivation of each gene at the *INK4b/ARF/INK4a* locus will reveal their independent roles in development of MPNST and other sarcomas.

## 3. CDK Pathway Alterations in Prevalent Soft Tissue and Bone Sarcomas

The following sections summarize notable CDK and CDK pathway alterations associated with the more common adult and childhood subtypes of sarcoma. Importantly, the tumor types discussed below are not necessarily limited to adults or children but instead tend to be more prevalent in one age group versus the other. For instance, children can develop MPNSTs, synovial sarcomas, and chondrosarcomas even though those cancers are more common in adults, while adults can present with typical pediatric sarcomas, such as rhabdomyosarcoma, Ewing sarcoma, and osteosarcoma. A consolidated listing of the genetic alterations characteristic of each sarcoma is provided in Table 1.

Notable overlap exists in several of the genetic alterations found within multiple sarcomas although there are unique genomic events that also distinguish each sarcoma. A recent analysis of genomic profiles and clinical outcomes in two independent datasets of diverse soft tissue sarcomas identified the most frequently altered genes shared by most sarcomas, namely *TP53*, *CDKN2A*, *RB1*, *NF1,* and *ATRX* [36]. Strikingly, the only gene whose alterations were associated with worse overall survival across all types of localized soft tissue sarcomas was *CDKN2A*. These results suggest there is broad biological importance of the p16^INK4a^-CDK4/6-RB1 pathway and/or ARF signaling pathways in sarcoma pathogenesis, bolstering emerging evidence supporting the use of FDA-approved, CDK-targeted therapies for sarcoma treatment.

## 4. CDK Pathway Alterations in Common Adult Sarcomas

### 4.1. Undifferentiated Pleomorphic Sarcoma (UPS)

UPS (formerly called malignant fibrous histiocytoma) accounts for 15–20% of all soft tissue sarcomas (STS) and normally occurs in the limbs and trunk of adults >40 years of age [37]. UPS falls into the category of “a complex karyotype sarcoma” and is characterized by a lack of a line of differentiation. Loss of regions within chr 13q is the most frequent genomic alteration (up to 78% of tumors), which leads to deletion of the *RB1* gene located on chr13q14.2 [38,39]. Other RB1 pathway alterations leading to functional loss of RB1 activity are also common in UPS. Genetic mutation, deletion, or silencing of chr9p21, the region containing *CDKN2A*, result in loss of p16^INK4a^ and unrestricted activation of cyclin D-CDK4/6 kinases. Gains in chr12q13-15, a region that contains *MDM2* and *CDK4* genes, has been found in up to 30% of UPS [40]. As noted above, the MDM2 (mouse double minute-2) oncoprotein ubiquitinates the p53 tumor suppressor and promotes its proteasomal degradation. Since the CDK inhibitor, p21, is a transcriptional target of p53, overexpression of MDM2 causes downregulation of p21 and hyperactivation of CDKs. A similar outcome can be achieved by deletion and mutation of *TP53*, which has been observed in roughly 40% of UPS [41]. Recently, a unique bioluminescent mouse model of UPS was developed using viral-Cre recombinase to induce loss of floxed *Trp53* and *Pten* genes [42]. UPS formed in all mice following intramuscular or subcutaneous injection of virus. 

Activation of Ras signaling, which results in excessive MAPK activity and consequent increased transcription of cyclin D1 [43,44] and CDK4/6 [45,46] has also been linked to UPS development [47,48]. In a study containing 37 UPS patients, >80% displayed activated Ras/MAPK and PI3K/mTOR pathways [47]. Notably, all but one of the tumors that were analyzed for mutations expressed wildtype *NRAS*, *BRAF*, and *PIK3CA*. Thus, while oncogenic mutations in these pathway genes are uncommon, pathway hyperactivation is frequent and predicts poor recurrence free and overall survival. This prompted development of a conditional *Kras^G12D/+;p53flox/flox^* genetically engineered mouse model (GEMM). These double mutant mice develop UPS with 100% penetrance and accurately mimic the gene expression signature and lung metastatic features of the human disease [49]. As p53-mediated tumor suppression is linked to upregulation of ARF (the alternative product of the *Cdkn2a* gene locus) by hyperactivated oncogenes like Ras [50,51,52], and mice lacking ARF primarily develop undifferentiated sarcomas [53], it was not surprising when Kirsch et al. demonstrated that conditional *Kras^G12D/+;Cdkn2aflox/flox^* mice efficiently develop UPS [54]. Other studies showed that intramuscular targeting of the Neurofibromatosis Type 1 (*Nf1*) gene, which encodes a negative regulator of Ras, combined with deletion of *Cdkn2a* leads to UPS formation [55]. These studies highlight the biological importance of excessive Ras signaling associated with CDK dysregulation in UPS biology.

### 4.2. Myxofibrosarcoma (MFS)

Until recently, myxofibrosarcoma was grouped among UPS but changes in morphologic and immunohistochemical criteria prompted their separation into distinct categories. Roughly 5% of sarcoma cases are classified as MFS, which is characterized by myxoid histology with hypocellular appearance and complex karyotypes [56]. MFS lesions display highly complex karyotypes that were recently deemed genetically indistinguishable from UPS tumors [57]. Although the two subtypes are genetically similar, MFS with at least 10% myxoid area generally display better prognosis than UPS. MFS lesions with less than 10% myxoid area display prognosis similar to that of UPS [58]. Further, MFS and UPS display differential locations. MFS are typically superficial and located in subcutaneous tissue (64.9%) whereas UPS lesions are almost always deep-seated below the muscle fascia (92.3%). Recent whole-exome sequencing of 99 MFS tumors revealed frequent alterations in genes related to the p53 and RB1 tumor suppressor pathways, including *TP53*, *MDM2*, *RB1*, *CDKN2A/B*, *CCND1* (cyclin D1), and *CDK6* [59]. CDK6 overexpression in MFS, primarily driven by gene amplification on chr 7q, is associated with poor patient outcomes [60]. CDK6 shares 70% amino acid identity with CDK4. As such, the two proteins are for the most part considered functionally interchangeable although they often display differential, tissue specific expression [61,62]. Additional mutations that have been observed in MFS include those causing inactivation of the *NF1* gene, which causes heightened Ras signaling and upregulation of *CCND1* transcription [63,64]. Increased cyclin D1 has been shown to drive tumorigenesis in NF1 mutant/Ras-activated cancers [43,44].

### 4.3. Liposarcoma (LPS)

Liposarcoma arises from adipocytes and accounts for a significant proportion (~13%) of sarcoma diagnoses. LPS is subcategorized into three groups based on molecular profiles and growth behavior: (1) Well- and de-differentiated (WD/DDLPS) – *CDK4* and *MDM2* amplification; (2) myxoid/round cell (M/RCLPS) – t(12;16)(q13;p111) translocation creating a TLS-CHOP fusion protein; and (3) pleomorphic (PLPS) – complex karyotype with consistent loss of p53 and RB1 [65]. Roughly 60% of LPS cases are WD/DDLPS, while PLPS is the rarest (~5%) [66].

WD and DDLPS subtypes are distinguished by differences in growth and genetic complexity. In WDLPS, the initiating event driving tumor formation is amplification of the *MDM2* gene. These tumors carry the potential for local recurrence. Additional genomic alterations cause tumor progression into DDLPS, which in turn leads to increased risk for metastasis and death [65]. The translocation that occurs in M/RCLPS produces a fusion protein containing the amino terminus of translocated in liposarcoma (TLS) to a full-length transcription factor, C/EBP homologous protein (CHOP). This translocation is also commonly referred to as FUS-DDIT3, where FUS (fused in sarcoma) corresponds to TLS and DDIT3 (DNA damage inducible transcript 3) represents CHOP [67]. Normally, CHOP is induced during stress conditions and causes growth arrest; however, the fusion of TLS prevents its normal function [68]. TLS-CHOP causes abnormal expression of G1 phase regulators, such as cyclin D1, cyclin E, CDK4, and CDK2 [69], and its constitutive expression in mice is sufficient to drive M/RCLPS formation with 100% penetrance [70]. PLPS are similar to UPS at the genomic level, where multiple alterations occur and *RB1* is frequently lost. However, unlike UPS, these tumors are not known to have amplification in *MDM2* [37,71].

### 4.4. Leiomyosarcoma (LMS)

Leiomyosarcoma are neoplasms of smooth muscle origin that comprise 10–20% of sarcoma diagnoses. LMS fall into the “complex karyotype” category of sarcomas with numerous, typically non-recurrent, chromosomal alterations [72]. One of the most frequent disruptions in LMS is hyperactivation of the PI3K-Akt-mTOR pathway, whether through deletion of *PTEN* or amplification/upregulation of IGF1R, AKT, RICTOR, and MTOR [57]. Notably, genetic deletion of *Pten* in smooth muscle is sufficient to drive LMS development in mice [73]. Targeted exome sequencing in tumors also shows loss of *RB1* in 54% of LMS [74] with alterations in other RB1 pathway components (*CDKN2A*, *CCND1*, *CCND3*) observed in up to 90% of LMS patients [75]. In addition, deletion and mutation of *TP53* occurs frequently in roughly 50% of cases. Deletions and promoter methylation of the *CDKN2A* gene as well as amplification of *MYC* have been observed in patient LMS specimens [76].

### 4.5. Malignant Peripheral Nerve Sheath Tumors (MPNSTs)

MPNSTs are a highly aggressive, deadly subtype of sarcoma that accounts for 5–8% of diagnoses. These tumors predominantly occur in the extremities and arise from myelinating Schwann cells that surround neuronal axons. MPNSTs can develop sporadically (50%), most commonly in middle to advanced aged patients, or in association with NF1, a neurological cancer predisposition syndrome [38]. NF1 is an autosomal dominant condition that occurs through germline loss-of-function mutation of the neurofibromin 1 gene (*NF1*), which as noted above ultimately results in hyperactivation of Ras [77]. NF1 patients display a mosaic of clinical manifestations, most notably including the development of benign lesions called neurofibromas (NFs). NFs can be small, dermal nodules located exclusively in the skin or much larger, deep-seated tumors termed plexiform neurofibromas (PNFs) that form around a nerve plexus or multiple nerve bundles. Approximately one-third of PNFs transform into MPNSTs, the major cause of death in NF1 patients [78,79].

Both sporadic and NF1-associated MPNSTs have complex karyotypes characterized by *NF1*-inactivating mutations along with frequent genetic disruptions of *CDKN2A* and polycomb repressor complex-2 (PRC2) components, *SUZ12* or *EED* [78,80]. Inactivation of PRC2, which causes loss of histone H3 lysine 27 (H3K27) tri-methylation and global changes in gene expression, is a marker of poor survival in MPNST [77]. Normally, PRC2 transcriptionally represses several genes in the CDK pathway, including *CDKN2A* and the genes encoding cyclins D1 and E1, *CCND1* and *CCNE1* [81]. Since most MPNSTs lack *CDKN2A* because of early genomic deletion of the locus in pre-malignant lesions, the de-repression of *CCND1* and *CCNE1* upon PRC2 loss would be expected to drive MPNST progression by abolishing RB1 activity. Additional CDK pathway alterations commonly seen in MPNSTs include inactivating mutations of *TP53*, loss or mis-localization of the CDK inhibitor, p27^KIP1^, and amplification or increased expression of cyclin E1 [78,80,82,83]. One study showed that cytoplasmic p27^KIP1^ localization correlates with increased nuclear cyclin E1 expression and poor prognosis in MPNST patients [83]. Most recently, immunohistochemical analyses of patient matched PNFs and MPNSTs revealed dramatic upregulation of an oncogenic GTPase, RABL6A, in MPNSTs [84]. RABL6A is a newly recognized negative regulator of RB1 and p53 signaling [85,86,87]. In MPNSTs, RABL6A was found to promote MPNST pathogenesis, in part, through its downregulation of p27^KIP1^, activation of CDKs, and inhibition of RB1 [84].

The stepwise nature of genetic alterations that occur during MPNST development in NF1 patients has informed the successful generation of multiple mouse models of the disease. The first MPNST GEMM combined *Nf1* and *Trp53* linked germline mutations [88], while subsequent work showed that combined loss of *Nf1* with either *Cdkn2a* or *Pten* is also sufficient to generate MPNSTs [89,90,91]. MPNST formation may not require *Nf1* loss as a sporadic GEMM has been developed through *Pten* loss plus *EGFR* overexpression, but the study did not determine if resulting tumors had sustained spontaneous *Nf1* alterations [92]. Spatial and temporal control of MPNST development has been achieved by injecting adenoviral Cre recombinase into the sciatic nerve of *Nf1^flox/flox^; Cdkn2a^flox/flox^* mice [55]. Alternatively, adenoviral CRISPR-Cas9 gene editing of *Nf1* with *Trp53* in the sciatic nerve has generated MPNSTs with complete penetrance in non-genetically altered mice [93]. The latest MPNST GEMM has shown a role for *ARF* loss in MPNST development. Specifically, conditional loss of *Arf* alone in an *Nf1*-associated mouse PNF model was found to drive their transformation into transitional lesions called atypical neurofibromatous neoplasms of uncertain biological potential (ANNUBPs), some of which subsequently progress to MPNST [94]. Together, these mouse models have enhanced our understanding of MPNST biology and provided outstanding platforms for testing new therapies.

### 4.6. Synovial Sarcoma (SS)

Synovial sarcoma primarily arises in young adults (mean age 35 years) from mesenchymal progenitor cells. These tumors have few mutations other than the primary oncogenic translocation, t(X;18), which is necessary and sufficient to drive their formation [95]. This translocation creates a fusion protein of the C-terminus of synovial sarcoma translocation, chr 18 (*SS18*) to the C-terminus of synovial sarcoma, X (*SSX*). Normally, SS18 incorporates into the chromatin remodeling complex, mammalian switch/sucrose non-fermentable (SWI/SNF), which enables access to DNA for transcription. SSX is associated with transcriptional repression among the polycomb repressive complex (PRC) proteins [96]. The fusion protein enhances the ability of SSX to interact with transducing-like enhancer of split 1 (TLE1), a transcriptional corepressor. Together, these factors are capable of activating (SS18) and repressing (SSX) transcription, thereby disrupting both aspects of epigenetic control. This interaction facilitates repression of *CDKN2A* [97,98]. However, the full extent of epigenetic dysregulation mediated by SS18-SSX has not been elucidated.

Taking advantage of the simple karyotype of SS, development of a SS mouse model was achieved by expressing this fusion protein in immature myoblasts. Tumors generated in this mouse model showed increased expression of CDK4 as well as multiple cyclins (D1, B1, A2, I, and F), strongly implicating a role for CDKs in driving SS formation [99]. A recent study investigated CDK4 expression in SS cell lines as well as a tissue microarray (TMA) containing 50 patient cases [96]. All four cell lines (SYO-1, Yamato, Fugi, and Aska) exhibited high levels of CDK4, while only SYO-1 and Yamato displayed CDK6 expression. Importantly, all four cell lines displayed RB1 hyper-phosphorylation indicative of its inactivation. Further, the TMA revealed a significant correlation with clinical stage and CDK4 expression with elevated CDK4 levels observed in higher grade lesions. The status of RB1 phosphorylation and CDK6 expression was not reported for the TMA samples [100].

### 4.7. Chondrosarcoma (CS)

Chondrosarcomas are neoplasms of the bone, where tumor cells produce excess cartilaginous matrix. Almost all chondrosarcomas, with a few exceptions, arise after the age of 40 with tumors primarily affecting the pelvis, femur, humerus, and ribs [101]. CS is divided into three subtypes, where the majority (85%) are conventional CS. The other two subtypes (~2% mesenchymal and ~10% dedifferentiated) are more aggressive, higher grade lesions. The most common mutations in conventional chondrosarcoma are found in isocitrate dehydrogenase genes (*IDH*), which produce enzymes that catalyze the conversion of isocitrate to alpha-ketoglutarate [102]. Alterations in the *TP53* gene are also common and correlate with histological grade. Overexpression of MDM2 has also been observed. Further, the RB1 pathway is often disrupted via enhanced activity of the CDKs, inactivation of their inhibitors (such as p16^INK4a^), or loss of the *RB1* gene. It is estimated that ~96% of high-grade CS have alterations in the RB1 pathway, whether through loss of *CDKN2A*, overexpression of CDK4 or cyclin D1, or loss of *RB1* [101].

## 5. CDK Pathway Alterations in Common Childhood and Adolescent Sarcomas

### 5.1. Rhabdomyosarcoma (RMS)

Rhabdomyosarcoma arises from the skeletal muscle lineage and is the most common soft-tissue sarcoma in children, accounting for 4.5% of all pediatric cancers. RMS is divided into two subtypes: embryonal (ERMS) for young patients (2–6 years) or alveolar (ARMS) for older patients (10–18 years). ERMS generally develops in the head/neck and genitourinary areas while ARMS arises in the trunk and extremities [103]. ERMS typically has a good prognosis, and the most recurrent genetic alteration is loss of heterozygosity at chr 11p15 locus. This region contains *CDKN1C*, which encodes the CDK inhibitor p57KIP2, as well as the long noncoding RNA *H19* and *IGF2* [104].

ARMS is more aggressive than ERMS and can be divided into two genotypes based on the presence/absence of *PAX3-FOXO1A* or *PAX7-FOXO1A* gene fusions encoded by the t(2;13) and t(1;13) chromosomal translocations, respectively. In fusion-positive ARMS, the mutational burden is low with the most common alterations being gene amplifications of chr 2p24 (containing *MYC*) and chr 12q13-q14 (containing *CDK4*). Myc is a powerful, oncogenic transcription factor whose overexpression directly increases the transcription of many CDK pathway genes, including *CDK4*, *CDK6*, *CDC25A*, and *SKP2*, among others [105,106]. Skp2, an F-box protein and key component of the SCF ubiquitin ligase, is upregulated in many cancers and acts in part by targeting p27^KIP1^ (as well as the other CIP/KIP proteins) for proteasome degradation [107,108,109,110]. Tumors with amplifications in the CDK4 region are associated with worse overall survival. The PAX3-FOXO1A fusion protein has been shown to reduce p27^KIP1^ protein levels via elevated expression of Skp2 and enhanced proteasomal activity [111].

In contrast, fusion-negative ARMS exhibit complex karyotypes with numerous genetic alterations. The most common mutations occur in the Ras oncogenic pathway; either in one of the Ras genes (*NRAS, HRAS, KRAS*) or its effectors, *BRAF* and *PIK3CA*. Additionally, focal losses in *TP53*, *NF1*, and *CDK2NA* are frequently observed within tumors as well as copy number gains in chr 12q14-15 (containing *MDM2* and *CDK4*) [112]. A recent study revealed elevated levels of the transcription factor Forkhead Box F1 (FoxF1) in ARMS, promoting tumorigenesis by repressing expression of p21^CIP1^ [113]. Efforts to develop a mouse model of ARMS demonstrated that the *PAX3-FOXO1A* fusion protein is not sufficient to drive tumor development, rather second hits of *Trp53* or *Ink4a/Arf* are necessary [49,114].

### 5.2. Osteosarcoma (OS)

Osteosarcoma is the most common primary bone tumor in adolescents, arising mainly in the long bones of the limbs. OS is characterized by the aberrant production of immature bone by mesenchymal cells. Sadly, patient survival has not increased since advances in treatment in the 1990s, where the 10-year survival remains at roughly 70% [115,116].

These tumors display complex karyotypes with high chromosomal instability. Deletion, loss of heterozygosity, and mutations arise frequently in CDK-related tumor suppressor genes (*TP53, RB1, CDKN2A, PTEN*) as well as amplifications of oncogenes (*CDK4, MDM2, MYC, TWIST1, CCND3, CCNE1*). *TWIST1* can negatively regulate p16^INK4A^ and p21^CIP1^ by preventing their Ras- and p53-mediated promoter activation, linking *TWIST1* to aberrant CDK activity [117]. The majority of OS cases display inactivating mutations in *TP53* (80%) and *RB1* (70%) genes [116,118,119]. Given the significant role of RB1 and p53 loss in OS genesis, inherited retinoblastoma and Li-Fraumeni syndrome patients bearing germline mutations in those genes are at significantly higher risk of developing OS [120]. Indeed, genetically engineered mouse and pig models with inactivating mutations in *RB1* and/or *TP53* provide useful in vivo models of the disease. For example, dual inactivation of *RB1* and *TP53* in osteoblast precursors in mice caused early onset OS development [121] while homozygous mutation of *TP53* in Yucatan mini-pigs yielded cranial OS tumors with altered expression of mutant p53 target genes, including upregulation of cyclin B1 [122].

### 5.3. Ewing Sarcoma (EwS)

Ewing sarcoma is a small round cell tumor that is believed to arise from either neural crest or mesenchymal stem cells. Roughly 70% of EwS occurs in the bone, particularly the long bones, otherwise tumors form in the soft tissue (nerves or cartilage) surrounding the bone. It often develops when bones are growing rapidly during puberty, so it usually affects people in the second decade of life (10–19 years) [123]. EwS has a 70% five-year survival rate for localized disease, although that drops for patients with metastatic tumors to 15 to 30% five-year survival [124]. Chemotherapy is a critical part of managing EwS as it is used for all patients in the initial therapy to shrink the primary tumor [125].

EwS typically exhibit simple karyotypes with a single driving mutation [126]. In roughly 85% of cases, a chromosomal translocation between chr 22 and chr 11 is present, resulting in the fusion of the *EWSR1* (Ewing sarcoma breakpoint region 1) and *FLI1* (Friend leukemia integration 1) genes. The fusion protein acts as an aberrant transcription factor by converting microsatellites into active enhancer regions; thereby, promoting the expression of oncogenes [127]. Recent evidence shows cell-to-cell heterogeneity of EWSR1-FLI1 levels in tumors facilitate different cellular processes, where low levels promote migration and invasiveness and high levels enable proliferation [128]. In addition, the fusion protein has been shown to upregulate c-Myc and conversely downregulate the CDK inhibitor, p57^KIP2^ [129]. Cooperating mutations in *TP53*, *RB1*, and *CDKN2A* as well as overexpression of cyclin D1 are observed [130]. Most efforts to develop a mouse model of EwS have failed to recapitulate the disease as the choice of target tissue for fusion protein expression is critical. For example, expressing the *EWS-FLI1* gene fusion in hematopoietic cells induces leukemias rather than sarcomas [131]. Expressing *EWS-FLI1* in primitive mesenchymal tissues in combination with *Trp53* deletion results in poorly differentiated soft tissue sarcomas [132]; however, a true EwS mouse model is yet to be developed [49].

## 6. Transcriptional and “Other” CDKs

The following section summarizes what is known about the transcriptional and “other” CDKs in sarcoma. Since our understanding is incomplete, Table 2 briefly describes the functions of these CDKs and the other types of cancer in which they play a role. These studies reveal significant contributions of non-cell cycle CDKs in human cancers, warranting greater investigation into their roles in sarcoma development and progression.

While CDKs and their related pathways components involved in cell cycle control are frequently altered in most types of sarcoma, our understanding of the transcriptional (7, 8, 9, 12, 13, and 19) and non-canonical “other” (5, 10, 11, 14–18, and 20) CDKs is significantly lacking. Transcriptional CDKs are critical to multiple processes of the transcription cycle, including RNA capping, splicing, 3’ end formation, and export. “Other” CDKs function in numerous cellular activities, many of which are not fully understood and vary based on context and cell type. Of the aforementioned sarcoma subtypes, only a few studies have provided insight into the role of non-cell cycle CDKs in their pathogenesis.

CDK9 regulates transcription by acting as the catalytic subunit of the positive transcription elongation factor b (P-TEFb), phosphorylating the C-terminal domain (CTD) of RNA polymerase II (RNA pol II) and promoting transcription elongation. Numerous studies have shown the importance of hyperactive CDK9 in the development of many human cancers—including hematologic, breast, liver, lung, and pancreatic tumors. A common mechanism by which CDK9 promotes aberrant cell proliferation is through increased transcription of MYC [140]. Recent evidence implicates CDK9 in sarcomagenesis, particularly in OS and SS. In a TMA containing samples from 59 sarcoma patients, roughly 60% exhibited high CDK9 expression which correlated with worse patient survival. In agreement with those results, knockdown or pharmacological inhibition of CDK9 in SS cells caused cell death, growth arrest, and reduced motility [141]. Likewise, increased expression of CDK9 was observed in osteosarcoma cells lines compared to normal osteoblasts and ~67% of osteosarcoma patient tumors (from a TMA containing 70 OS) had elevated CDK9 [142]. High expression of CDK9 again correlated with worse overall survival. These studies establish the importance of CDK9 to OS and SS progression.

CDK11 is classified as an “other” CDK as it functions in multiple cellular processes. It is encoded by two genes (CDC2L1 and CDC2L2) and produces three different isoforms, each with its own independent functions [6]. CDK11p46 exhibits a pro-apoptotic role, CDK11p58 functions during mitosis, promoting centrosome maturation and mitotic spindle assembly, and CDK11p110 regulates transcription and RNA splicing [143]. Overexpression and heightened activity of CDK11 has been noted in several cancers including breast, multiple myeloma, colon, and cervical cancer [6]. Additionally, CDK11 has been linked to both osteosarcoma and liposarcoma cell proliferation and survival [144,145,146]. It was identified in a kinome-wide, shRNA screen in OS cells, where its knockdown caused significant growth inhibition. Follow-up revealed CDK11 is highly expressed in OS cell lines and patient tumor samples (*N* = 45), where its elevated expression correlated with poor survival. Recent work by the same group showed increased CDK11 expression in LPS specimens compared to benign lipomas (*N* = 41). Both studies revealed CDK11 is critical for the growth and survival of OS and LPS cell lines. In LPS cells, knockdown of CDK11 correlated with a decrease in anti-apoptotic factors. The role and importance of the other molecular mechanisms controlled by CDK11 are less clear in these tumors, meriting additional investigation.

CDK14 (previously PFTK1) falls into the “other” category of CDKs. It contains a “TAIRE” amino acid motif that defines a subfamily comprised of CDKs 14-18, all of which have been understudied. The biological function of CDK14 is not fully understood although it has been shown to phosphorylate RB1 (the exact residues remain unknown) and is linked to activation of the Wnt pathway [147,148]. Increased expression of CDK14 has been implicated in colorectal cancer and is associated with poor prognosis [149]. Recent evidence suggests a role for CDK14 in the pathogenesis of OS. Its expression was significantly higher in OS tissue versus adjacent, nontumor tissue (*N* = 91) and higher levels correlated with increased tumor size and worse histological grade as well as poor overall and disease-free survival [150]. This study also showed that microRNA (miR)-216a inhibits the expression of CDK14, preventing OS cell proliferation and migration. While much more remains to be learned about the regulation and activity of CDK14 in OS and other cancers, current evidence suggests CDK14 is a driver of OS pathogenesis.

## 7. CDK Pathway Alterations—Are They Drivers or Passengers in Sarcoma Development?

As normal cells sustain DNA damage and evolve into malignant tumors, they undergo numerous genetic and epigenetic alterations that promote their transformation. Most cancers display two to eight mutations that provide a selective growth advantage; these are classified as “driver” mutations [179]. In contrast, other changes in the tumor genome and epigenome that are dispensable for the tumor phenotype (i.e., have no biological consequence) are dubbed “passenger” mutations. Driver genes, which can be defined as those bearing specific alterations (e.g., point mutation, amplification, translocation, deletion or epigenetic modifications) that critically promote tumorigenesis in a significant percentage of tumors, have been categorized into major signaling pathways required for cell fate, survival, and genome maintenance. Each driver mutation alone is thought to provide a small growth advantage to the cell, but when that increased rate of growth accumulates over years and is compounded by the effects of other driver mutations, large neoplasms can result.

Driver events may initiate tumor development or occur later in the neoplastic cascade to enforce malignant progression. The well-studied molecular pathogenesis of pancreatic ductal adenocarcinoma (PDAC) effectively illustrates that concept [180,181]. Activating *KRAS* mutations occur in over 90% of PDAC lesions of all grades whereas inactivation of *CDKN2A, p53,* and *SMAD4* is seen with increasing frequency in type 2 and 3 pancreatic intraepithelial neoplasms. Those findings, supported by biological evidence from genetically engineered mouse models, suggest *KRAS* alterations drive PDAC initiation while subsequent mutations in *CDKN2A* and other tumor suppressors are rate-limiting for tumor progression [181]. Each driver mutation, whether it is required for tumor initiation or progression, is a potentially actionable event in cancer therapy. Ras has traditionally proven challenging to target pharmacologically, although a unique inhibitor of the KRasG12C mutant (prevalent in lung cancer, rare in PDAC) is currently in clinical trials (NCT03681483, NCT04185883). Recently, there has been rising interest in targeting MEK and CDK4/6 for PDAC treatment as mutant KRas promotes MEK activation and *CDKN2A* loss causes CDK4/6 activation. Results from preclinical models have been promising [182].

Are alterations in CDKs and CDK pathways driving events in sarcoma pathogenesis, as they are in PDAC? The preponderance of data from molecular profiling of human tumors combined with functional studies in pre-clinical sarcoma models (see Table 1 and Table 3) would suggest the answer is yes for many types of sarcoma. In some cases, CDK alterations drive sarcoma initiation while in others they are important for malignant progression. For example, the vast majority of WD/DDLPS lesions harbor amplification of *CDK4* as part of the chr 12q-13-15 amplicon, which occurs early in their genesis and provides a clear selective growth advantage [57]. Similarly, multiple sarcoma subtypes, including UPS and MPNST, disrupt *CDKN2A* in early stages of their development to promote unrestricted growth and cellular transformation. By comparison, in certain sarcomas such as OS and EwS, *CDKN2A* loss is observed later in tumor genesis, suggesting that CDK activation in those tumors may promote progression rather than initiation. Results from pre-clinical studies (see Section 8) evaluating the effects of CDK inhibition in OS and EwS models support this idea since both tumor types depend on CDK activity for proliferation and survival.

It is less clear for some sarcomas if CDK pathway dysregulation is a driving event essential to their development. MFS is an excellent case in point. MFS lesions have highly complex karyotypes with frequent alterations in many well-established tumor suppressor genes and oncogenes within the CDK pathways, including *RB1*, *CDKN2A/B*, *CCND1*, and *CDK6* to name a few. This makes it difficult to discern driving mutations from passenger mutations. While *CDK6* amplification and overexpression is observed in about 25% of MFS and correlates with poor patient prognosis [60], more extensive genomic analyses of patient samples at different stages of tumor development paired with functional studies in pre-clinical models are needed to determine if *CDK6* is a true driver of MFS.

A key reason for defining driver mutations in tumors is to identify the most relevant, actionable targets to improve available therapies for patients, which for sarcoma are severely lacking. We suggest two basic criteria should be met to justify the use of CDK targeted agents for sarcoma therapy. First, does molecular genetic profiling adequately predict or establish CDK activation in the patient tumor? Second, is there compelling pre-clinical data demonstrating a clear tumor suppressive benefit of CDK inactivation in that subtype of sarcoma? The next section addresses the latter issue and reveals growing evidence that CDK-targeted therapy may have clinical efficacy against multiple subtypes of sarcoma. As we will discuss, the success of CDK4/6 inhibitors in patients with dedifferentiated liposarcoma has validated *CDK4* gene amplification as a driver mutation in this subtype and is leading the way for similar trials in additional soft tissue sarcoma subtypes.

## 8. CDK-Targeted Anti-Cancer Therapy

The following section summarizes the use of CDK-targeted therapy with an emphasis on its utility in treating sarcoma. Toward that end, Table 3 highlights a growing list of pre-clinical studies and clinical trials in sarcoma that employ agents targeting the CDKs.

Soon after the discovery of mammalian CDKs in the early 1990s, the idea of pharmacologically targeting the cell cycle CDKs for anti-cancer therapy emerged. High expectations for first generation CDK inhibitors were unfortunately not realized as those agents performed poorly in patients, displaying more toxicity than anti-tumor activity. A major problem with those early CDK inhibitors, such as flavopiridol and roscovitine, was their lack of specificity for particular CDKs. As competitive inhibitors with ATP, they bound to the ATP pocket of many different CDKs (and even some other kinases, like EGFR and PKA), thereby acting as pan-CDK inhibitors. The breadth of their activity (e.g., flavopiridol effectively inhibits CDKs 1, 2, 4, 6, 7 and 9) correlated with several dose-limiting toxicities in patients including diarrhea, nausea/vomiting, fatigue, and myelosuppression.

Second generation inhibitors generally target fewer CDKs but still act as multi-CDK inhibitors (e.g., dinaciclib strongly inhibits CDKs 1, 2, 5, and 9). These inhibitors compete with ATP binding but also interact with unique regions outside of the ATP pocket, thus allowing for greater specificity. While many of these inhibitors displayed less severe toxicities, their anti-cancer activities in early phase clinical trials were highly variable between different tumor types [183,184]. A number of excellent, comprehensive reviews describe the evolution of CDK-targeted therapies and highlight the development of newer, more selective CDK targeted drugs [184,185].

Efforts are now aimed at developing molecules that better target select CDKs to maximize anti-tumor efficacy while diminishing toxic side effects. Currently, there are few single CDK-specific inhibitors available and even fewer whose effects have been studied extensively. For example, inhibitor design is no longer solely focused on targeting the ATP pocket; rather, new classes of inhibitors are aimed at altering/preventing protein–protein interactions or conformational changes. Time will tell if the efficiency of these targeted compounds will be superior to current CDK inhibitory agents [184]. Of the canonical inhibitors, the best performing agents are the selective inhibitors of CDK4/6 that include palbociclib, ribociclib, and abemaciclib. All are potent, orally bioavailable drugs with abemaciclib having the added ability of crossing the blood–brain barrier. There is tremendous excitement about the clinical benefits of these agents. Currently, all three CDK4/6 inhibitors are approved (in combination with aromatase inhibitors) for the treatment of estrogen receptor-positive/HER2-negative metastatic breast cancer [186,187,188,189,190], and many clinical trials using these drugs are in progress for other types of cancer [191,192,193,194,195,196,197].

The efficacy of CDK4/6 inhibitors relies on the presence of wild-type RB1 in the cancer cells as inhibition of CDK4/6 blocks proliferation through the accumulation of hypo-phosphorylated RB1 [189]. Numerous pre-clinical studies have demonstrated the beneficial effects of CDK4/6 inhibition for RB1-positive human cancers; however, drawbacks of this targeted therapy are coming to light. First, CDK4/6 inhibitors are largely cytostatic. Second, monotherapy with CDK4/6 inhibitors is often over-ridden by CDK2-mediated phosphorylation of RB1, which can bypass the need for CDK4/6 and drive resistance to CDK4/6 inhibitors [190,197]. Studies in MPNSTs highlight this concept, where sustained CDK4/6 inhibition leads to acquired drug resistance that can be blunted by concomitant CDK2 inhibition [84]. Importantly, tumor cells have numerous mechanisms of resistance to CDK4/6 inhibitors at their disposal, which are both cell cycle specific (e.g., *CCNE* amplification and CDK2 hyperactivation) and independent of the cell cycle (e.g., activation of PI3K/AKT/mTOR signaling) (reviewed in [198]).

Many investigations are exploring the benefits of pairing CDK inhibitors with drugs that indirectly activate RB1, for example, by decreasing the activity of the Ras-MEK-ERK pathway. The underlying rationale is based on observations that MEK-ERK signaling activates cyclin D-CDK4/6 by upregulating the expression of each kinase subunit while downregulating the CDK inhibitory Cip/Kip proteins. Consequently, MEK inhibitors synergize with CDK4/6 inhibitors since each drug antagonizes CDK4/6 through distinct mechanisms [197]. Ruscetti et al. found that combining the MEK inhibitor, trametinib, with the CDK4/6 inhibitor, palbociclib, effectively suppressed Ras-driven lung cancer [199]. Interestingly, the efficacy of that combination was greatly enhanced in immunocompetent mice since it induced a senescence-associated secretory phenotype (SASP) that promoted natural killer cell recruitment and cytotoxicity in the tumors. In pancreatic ductal adenocarcinoma, the same MEK-CDK4 inhibitor combination triggered SASP-dependent tumor vascularization that facilitated chemotherapy uptake as well as increased CD8+ T cell infiltration into tumors that potentiated PD-1 checkpoint blockade [182].

In sarcomas, the prevalence of CDK dysregulation has galvanized efforts investigating CDK-targeted therapy to combat disease progression. An increasing number of pre-clinical studies using CDK inhibitors, alone or combined with other targeted agents, have been performed in cultured sarcoma cells and mouse models. The top section of Table 3 lists those studies and provides a brief description of the treatment outcome. Many of the studies demonstrated that monotherapy targeting specific CDKs induces a reversible cell cycle arrest, as expected. Some studies, however, revealed additional benefits of CDK inhibition including downregulation of other CDKs [200,201], decreased migration [141,202], and enhanced efficacy of other targeted agents [203,204]. Among the rationale combination therapies evaluated, THZ531, a CDK12 inhibitor, was found to synergize with PARP inhibitors to prevent DNA damage repair and induce tumor cell death in EwS, significantly diminishing the tumor growth in mouse xenograft and PDX models of the disease [205]. Importantly, the studies listed in Table 3 reveal the versatility of CDK inhibitor combination therapies since agents acting on a variety of other targets/pathways are proving effective in the setting of CDK inactivation. For example, CDK inhibitors combined with drugs targeting MDM2 [206], BRD4 [207], receptor tyrosine kinases [208], and IGF1R [209] have been effective against various sarcomas.

It should be noted that some therapeutic strategies are designed to activate CDKs and thereby trigger tumor cell death through disruption of normal cell cycle checkpoints. These approaches rely on drugs that target kinase regulators of the CDKs, such as WEE1 kinase inhibitors, to induce premature activation of G2/M CDKs and unscheduled entry of S phase cells into mitosis [210]. Normally, WEE1 inhibits CDK1 and CDK2 activity by phosphorylating them at T14 and Y15 (Figure 1C), thus acting as a gatekeeper of the G2/M and DNA damage checkpoints. Disrupting those checkpoints with WEE1 inhibitors in cells that have incompletely replicated DNA or damaged DNA causes premature mitosis, leading to severe chromosomal abnormalities and aneuploidy that provoke mitotic catastrophe and death. Several studies are harnessing that information to explore novel combination therapies for sarcoma. One group showed that the WEE1 inhibitor, MK-1775, exacerbated the cytotoxic effects of a DNA replication stress drug, gemcitabine, in MPNST, UPS, and OS cell lines [211]. More recently, studies in EwS cells showed that inhibition of the ATR-CHK1 checkpoint pathway activates CDK2, which enhances DNA replication stress by targeting a subunit of the ribonucleotide reductase enzyme for proteasomal degradation [212]. Those findings led to concurrent targeting of ATR-CHK1 and WEE1 pathways, which killed EwS cells in a highly synergistic fashion. Genetic context, however, may dictate the use of CDK2 activation versus inhibition therapies. Specifically, Musa et al. found that EwS patients with elevated expression of the oncogenic transcription factor, MYBL2, have poor overall survival and that high MYBL2 levels sensitize tumor cells to anti-CDK2 therapy [213].

These and other encouraging results from pre-clinical studies involving CDK-targeted agents, the vast majority of which antagonize CDK activity, have warranted their clinical use for treating sarcoma patients (Table 3, bottom). Early phase studies in select soft tissue sarcoma subtypes are showing promising results, particularly for liposarcoma where there is frequent *CDK4* amplification. In a phase 2 study of patients with advanced or metastatic WD/DDLPS (NCT01209598), palbociclib therapy (125 mg for 21 days in a 28-day cycle) resulted in occasional tumor response along with a favorable progression-free survival rate of 57% at 12 weeks [214]. The most notable toxicity was neutropenia (grade 3), which was observed in 33% of patients, although this was less toxic than a previous dosing regimen (200 mg for 14 days). Currently, there is a multi-center phase 2 trial of palbociclib monotherapy in Spain for patients who have advanced sarcomas with elevated expression of CDK4 (NCT03242382). Moreover, CDK4/6 inhibitors such as palbociclib are recognized as high priority agents by the Children’s Oncology Group for testing in metastatic, relapsed Ewing sarcoma [215].

As shown in Table 3, most of the ongoing clinical trials for sarcoma have just started accruing patients and many involve combination therapy to prevent acquired resistance to CDK-targeted monotherapy. Lessons from the use of CDK-targeted therapies in other tumor types will hopefully guide emerging strategies for sarcoma. For instance, recent pre-clinical studies in pancreatic adenocarcinoma suggest that CDK4/6 inhibitors should be administered after (not before) cytotoxic chemotherapeutics since activation of RB1 represses the DNA repair machinery and enhances DNA damage-induced tumor cell death [216]. As our understanding of the CDKs expands and we learn more about their individual roles in sarcoma pathogenesis, it is fair to say these kinases represent increasingly valuable targets in the treatment of sarcomas.

## 9. Summary

Despite the diverse nature of sarcomas, activation of CDK pathways is a common alteration contributing to their pathogenesis. One of the more frequent changes is inactivation of the *CDKN2A* locus, resulting in loss of ARF-p53 and p16^INK4a^-RB1 tumor suppressive signaling and consequent hyperactivation of cell cycle CDKs. Loss of other CDK inhibitors, such as p27, and upregulation of cyclin partners are also predominant events leading to aberrant CDK activation in sarcomas. While more remains to be learned about the roles and significance of CDKs in the many different types of sarcomas, especially for CDKs with transcriptional or other activities besides cell cycle regulation, it is clear these kinases are key players in sarcoma biology. Continued studies of CDK dysfunction in sarcomagenesis are expected to solidify their importance in this disease and further justify CDK-based therapies for patients. Indeed, newer generation CDK inhibitors, particularly those targeting CDK4 and CDK6, are more specific and less toxic than earlier, more broadly acting compounds. Based on impressive anti-tumor activities in pre-clinical studies, CDK4/6 inhibitors have become a central component of current phase 1 and 2 clinical trials for various types of sarcoma. These drugs offer promising treatment options for sarcomas that generally lack effective systemic therapies. Because of the acquired drug resistance associated with CDK inhibitor monotherapy, combination therapy with other targeted agents or chemotherapeutics is expected to provide the greatest benefit to patients.

## Figures and Tables

**Figure 1 ijms-21-03018-f001:**
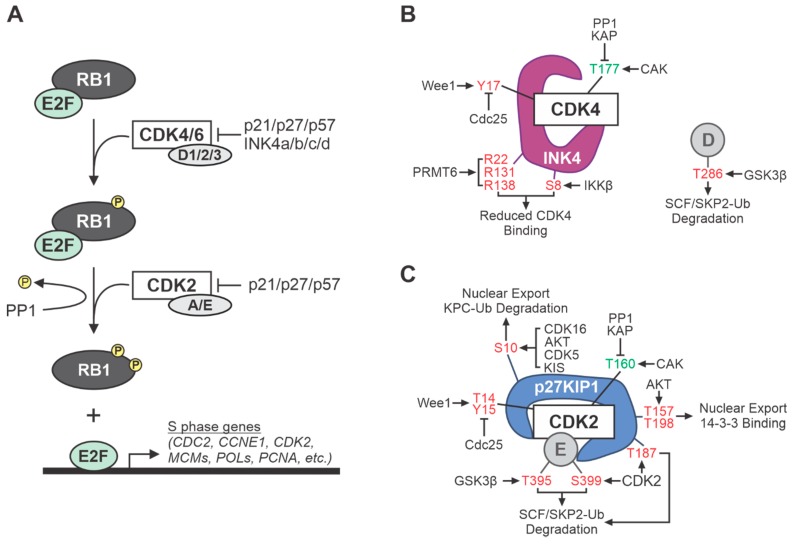
Overview of cell cycle cyclin-dependent kinase (CDK)-related signaling and regulation. (**A**) Several cell cycle CDKs inactivate RB1, promoting progression into S phase. RB1 binds and sequesters E2F transcription factors throughout G1. CDK4/6-cyclin D complexes phosphorylate RB1, priming it for subsequent phosphorylation by CDK2-cyclin A/E. Hyper-phosphorylation of RB1 causes release of E2F, enabling transcription of genes necessary for S phase and DNA synthesis. Two families of CDK inhibitors, CIP/KIP (p21/p27/p57) and INK4 (a/b/c/d), prevent RB1 phosphorylation; thereby halting cell cycle progression. (**B**) INK4 family members specifically inhibit CDK4 and CDK6 activity by preventing CDK-cyclin D complex formation. (**C**) CIP/KIP family members inhibit the various CDKs by binding the CDK-cyclin holoenzymes. Sites of post-translational modification in (**B**) CDK4, INK4, and cyclin D or (**C**) CDK2, p27KIP1, cyclin E that either promote (green) or inhibit (red) the function of each factor, as well as the modifying enzymes involved, are denoted.

**Table 1 ijms-21-03018-t001:** Genetic alterations of CDK pathway genes in sarcoma.

Gene	Protein	Alteration	Sarcoma Subtype
*RB1*	Retinoblastoma	Deletion, Mutation	UPS [36,38,39], MFS [36,59], PLPS [37,66], LMS [36,37,74], CS [101], OS [116,118], EwS [130], MPNST [82]
*CDKN2A*	p16^INK4a^ and ARF	Deletion, Mutation	UPS [36,40], MFS [36,59], LMS [75], MPNST [36,77,78,79,80,82,133,134,135,136,137,138], CS [101], ARMS [139], OS [116,118], EwS [130]
*CDKN2B*	p15^INK4b^	Deletion	MFS [36,59], MPNST [36,134,136]
*CCND1-3*	Cyclin D1-3	Amplification	MFS [59], LMS [75], CS [101], OS [116,118]
*CDK4*	CDK4	Amplification	UPS [40], WD/DDLPS [36,66], SS [100], CS [101], ARMS [111,112], OS [116,118]
*CDK6*	CDK6	Amplification	MFS [59,60]
*MDM2*	Mdm2	Amplification	UPS [40], MFS [59], WD/DDLPS [36,37,66], CS [101], ARMS [112,139], OS [116,118]
*TP53*	p53	Deletion, Mutation	UPS [36,41], MFS [36,59], PLPS [66], CS [101], ARMS [139], OS [116,118], EwS [130], MPNST [36,82], LMS [36,37,76]
*KRAS*	Ras	Amplification Mutation	UPS [47], ARMS [112]
*NF1*	Neurofibromin	Mutation	UPS [36], MFS [36,63], MPNST [36,77,78,79,80,82], ARMS [112,139]
*ATRX*	ATRX chromatin remodeler	Mutation	UPS [36], MFS [36], LPS [36]
*TLS*	Translocated in liposarcoma	translocation, (12;16)	M/RCLPS [66]
*CHOP*	C/EBP homologous protein		
*MYC*	Myc	Amplification	LMS [76], ARMS [111], OS [116,118], MPNST [134]
*PTEN*	Phosphatase and tensin homolog	Deletion	LMS [74], OS [116,118], MPNST [80,92]
*SUZ12*	Suppressor of zeste 12 protein homolog	Mutation	MPNST [78,79,80,82,133]
*EED*	Embryonic ectoderm development	Mutation	MPNST [78,79,80,82,133]
*SSX*	Synovial sarcoma, X	translocation, (X;18)	SS [95]
*SS18*	Synovial sarcoma translocation, chr18		
*IDH*	Isocitrate dehydrogenase	Mutation	CS [102]
*CDKN1C*	p57^KIP2^	Deletion	ERMS [104]
*PAX1*	Paired box 1	translocation, (2;13)	ARMS [112]
*FOXO1*	Forkhead box O1		
*BRAF*	B-Raf	Mutation	ARMS [112]
*PIK3CA*	p110α	Mutation	ARMS [112]
*TWIST1*	Twist family bHLH transcription factor 1	Amplification	OS [116,118]
*CCNE1*	Cyclin E1	Amplification	OS [116,118], MPNST [83]
*EWSR1*	Ewing sarcoma breakpoint region 1	translocation, (11;22)	EwS [127]
*FLI1*	Friend leukemia integration 1		

**Table 2 ijms-21-03018-t002:** Function and implications in human cancers of non-canonical CDKs.

**Transcriptional CDKs**			
**CDK**	**Function**	**Cancer**	**Ref**
CDK7	Subunit of TFIIH CDK-activating kinase	Hepatocellular carcinoma, breast, and gastric, glioblastoma	[5,151,152,153,154,155]
CDK8	Mediator complex Cyclin H inhibitory phosphorylation Direct interaction with NOTCH, TGF-β, Wnt, and STAT glycolysis	Colorectal, breast, pancreatic, melanoma	[156]
CDK9	Catalytic subunit of P-TEFb	Hematologic, breast, liver, lung, pancreatic, OS, SS	[140,141,142]
CDK12	Ser2 phosphorylation of CTD of RNA pol II RNA splicing Transcriptional termination 3’ end formation DNA damage response and repair	Breast, uterine, bladder	[157,158]
CDK13	Ser2 and Ser5 phosphorylation of CTD of RNA pol II	Hepatocellular carcinoma, colon, breast, gastric, melanoma	[159]
CDK19	Mediator complex	Prostate	[160,161]
**“Other” CDKs**			
**CDK**	**Function**	**Cancer**	**Ref**
CDK5	Neurite outgrowth and synaptogenesis Reduces insulin secretion RB1 phosphorylation DNA damage repair Cytoskeleton remodeling	Breast, lung, ovarian, prostate, neuroendocrine, multiple myeloma	[7,8,9,162,163,164,165]
CDK10	G2/M transition Promotes ETS2 degradation	Breast, prostate, gastro-intestinal, melanoma, hepatocellular carcinoma	[166,167,168,169]
CDK11	Apoptosis Mitosis Transcription/RNA splicing	Breast, multiple myeloma, colon, cervical, OS, LPS	[6,143,144,145,146]
CDK14	RB1 phosphorylation Wnt activation	Colorectal, OS	[149,150]
CDK16	Neuron outgrowth Spermatogenesis p27 phosphorylation	Non-small cell lung, breast, pancreatic	[170,171,172]
CDK20	G1-S transition Apoptosis Epigenetic control (EZH2-β-catenin-AKT signaling)	Glioma, hepatocellular carcinoma, colorectal, lung, ovarian, prostate	[173,174,175,176,177,178]

**Table 3 ijms-21-03018-t003:** CDK-targeted therapy in sarcoma.

**Pre-Clinical Studies**
**Sarcoma**	**Target**	**Drug Name**	**Outcome**	**Refs**
UPSOS	Pan-CDK	Flavopiridol	In vitro: growth inhibition, apoptosis	[217]
Drug-resistant OS and EwS	In vitro: apoptosisIn vivo: tumor growth inhibition	[218]
LPS	In vivo and Phase I Trial: Enhanced effects of doxorubicin	[203]
LMS	CDK2, 7, 9 and alkylating agent	Roscovitine + Cisplatin	In vitro: Roscovitine - G1 arrest, minimal apoptosis, decreased CDK2 expressionCombo: increased apoptosis	[200]
UPS, MPNST, LMS, MFS	CDK4/6	Palbociclib	In vitro: growth inhibition, senescenceIn vivo: variable response, dictated by CDK4 levels	[219]
OS and STS including LMS	CDK4/6 and WEE1 kinases	Palbociclib + AZD1775	In vitro: Palbociclib - reversible G1 arrest, but enhanced effects of S-G2 targeted agents	[204]
EwS	CDK12 and PARP	THZ531 + PARP inhibitors	THZ531: in vitro and in vivo cell cycle arrest, impaired DNA damage repairCombo: synergized to kill cells and tumors	[205]
EwS	CDK4/6 and IGF1R	Ribociclib or Palbociclib + AEW541	IGF1R induced an acquired resistance to CDK4/6 inhibitionCombo: synergistic cell cycle inhibition and non-apoptotic cell death in vitro and in vivo	[209]
EwS	USP7, MDM2/MDM4, and Wip1	P5091, ATSP-7041, and/or GSK2830371	In vitro: MDM2, MDM4, USP7 and/or PPM1D inhibitors (alone or in combination) - cytotoxic in p53-wild type EwS	[220]
LPS	CDK4/6	Ribociclib	In vitro: G0-G1 arrestIn vivo: LPS xenografts – halted tumor growth but acquired resistance	[221]
Fusion-positive RMS	CDK4/6	Ribociclib	In vitro: G1 phase arrest, diminished in tumors overexpressing CDK4In vivo: slowed tumor growth	[222]
SS	CDK9	LDC000067	In vitro: decreased cell number and viability, diminished 3D spheroid and colony formation as well as migration	[141]
OS	BRD4 + pan-CDK or CDK2	JQ1 + Flavopiridol or Dinaciclib	JQ1: induced apoptosis in vitro, suppressed in vivo growthCombo: enhanced cell death in vitro	[207]
Recurrent OS	RTKs + CDK4/6	Sorafenib + Palbociclib	Combo: greater PDX tumor inhibition than monotherapy and necrosis	[208]
CS	CDK4/6	Palbociclib	In vitro: decreased cell proliferation, migration, and invasion	[202]
CS	IL-1β	Diacerein	In vitro: decreased cell viability and proliferation, caused G2/M arrest with down-regulation of cyclin B1-CDK1 complex and CDK2 expression	[201]
OS	CDK2	Dinaciclib	In vitro: induced apoptosis at low nM concentrations	[223]
DDLPS	MDM2 + CDK4/6	RG7388 + Palbociclib	Combo: greater antitumor effects than monotherapy - in vitro: decreased cell viability, increased apoptosis – in vivo: decreased tumor growth rate and increased PFS	[206]
MPNST	CDK4/6 + CDK2	Palbociclib + Dinaciclib	Combo: low-dose combinations synergized in vitro and inhibited in vivo tumor growth	[84]
UPS, MPNST, OS	WEE1 + nucleoside analog	MK-1775 + Gemcitabine	MK-1775: forced entry into mitosis, enhanced effects of gemcitabineIn vivo: reduced osteosarcoma growth	[211]
EwS	CHK1 + WEE1	LY2603618 + AZD1775	In vitro: Combo induced activation of CDK1/2 and apoptosis not observed by either monotherapy	[212]
EwS	CDK2	CVT-313 and NU6140 (also inhibits AURKB)	In vitro: reduced growthIn vivo: NU6140 – reduced xenograft growthBoth responses dependent on MYBL2 expression	[213]
**Clinical Trials**
**Sarcoma**	**Target**	**Drug Name**	**Trial and Outcome**	**Refs**
Advanced/Metastatic LPS	CDK4/6	Palbociclib	NCT01209598: Phase 2, completed (Memorial Sloan Kettering Cancer Center)	[214]
Advanced sarcomas with CDK4 overexpression	CDK4/6	Palbociclib	NCT03242382: Phase 2, recruiting (multicenter trial Spain)	
Advanced bone (CS, OS, soft tissue sarcoma except LPS) with CDK pathway alteration	CDK4/6	Abemaciclib	NCT04040205: Phase 2, recruiting (Medical College of Wisconsin)	
Ewing Sarcoma	CDK4/6 + IGF1R	Palbociclib + Ganitumab	NCT04129151: Phase 2, recruiting (Dana Farber Cancer Institute)	
RB1 Positive Advanced Solid Tumors (including RMS and EwS)	CDK4/6	Palbociclib	NCT03526250: Phase 2, recruiting (National Cancer Institute)	
Recurrent/Refractory Solid Tumors (including EwS and RMS) in children	CDK4/6 + chemotherapy	Palbociclib + Temozolomide + Irinotecan	NCT03709680: Phase 1, recruiting (Pfizer)	
Advanced Solid Tumors	CDK4/6 + chemotherapy	Palbociclib + Cisplatin or Carboplatin	NCT02897375: Phase 1, recruiting (Emory University)	
Dedifferentiated LPS	CDK4/6	Abemaciclib	NCT02846987: Phase 2, recruiting (Memorial Sloan Kettering Cancer Center)	
Advanced/Metastatic Solid Tumors	Notch + CDK4/6	LY3039478 + Abemaciclib	NCT02784795: Phase 1, active (Eli Lilly and Company)	
Metastatic or Advanced, Unresectable STS	CDK4/6 + chemotherapy	Ribociclib + Doxorubicin	NCT03009201: Phase 1, not yet recruiting (OHSU Knight Cancer Institute)	
Advanced DDLPS and LMS	CDK4/6 + mTOR	Ribociclib + Everolimus	NCT03114527: Phase 2, recruiting (Fox Chase Cancer Center)	
Advanced WD/DDLPS	CDK4/6	Ribociclib	NCT03096912: Phase 2, recruiting (Assaf-Harofeh Medical Center)	
LPS	HDM2 + CDK4/6	HDM201 + Ribociclib	NCT02343172: Phase 1b/2, active (Novartis Pharmaceuticals)	
Advanced Solid Tumors with *CCNE1* Amplification	WEE1	Adavosertib	NCT03253679: Phase 2, recruiting (National Cancer Institute)	
Solid Tumors with Genetic Alterations in D-type Cyclins or CDK4/6 Amplification	CDK4/6	Abemaciclib	NCT03310879: Phase 2, recruiting (Dana-Farber Cancer Institute)	
Gastrointestinal Stromal Tumors (Refractory to Imatinib and Sunitinib)	CDK4/6	Palbociclib	NCT01907607: Phase 2, active (Institut Bergonié)	
Advanced WD/DDLPS	CDK4/6	Ribociclib	NCT02571829: Phase 2, unknown status (Hadassah Medical Organization)

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
