# Peer review of "CDKs in Sarcoma: Mediators of Disease and Emerging Therapeutic Targets"

_ijms, 2020, doi:10.3390/ijms21083018_

Round 1

Reviewer 1 Report

Major

UPS is an example not well taken. Despite the fact RAS may be involved in sarcomagenesis, this oncogene is rarely involved compared to several epithelial tumors. Thus, authors should shrink the paragraph to evidences related to clinical ground only. Mice models are always difficult to be translated into human sarcomas

Mixofibrosarcoma. Authors again span from CDK6 amplification (that is found in less than 25% of the cases) to RAS dysregulation. I suggest authors to be more focused because the paper is not on cell cycle alterations  but on CDK. Otherwise readers do not appreciate the true role of CDK in mixofibrosarcoma pathogenesis  (at most marginal) and might expect CDK inhibitor activity where none has previously been observed.

Author Response

We are grateful to both reviewers for their time and effort considering our manuscript. We believe the changes made in response to their insightful comments have strengthened and clarified the review, which we hope will be beneficial to the sarcoma and CDK signaling fields.

Reviewer 1

UPS is an example not well taken. Despite the fact RAS may be involved in sarcomagenesis, this oncogene is rarely involved compared to several epithelial tumors. Thus, authors should shrink the paragraph to evidences related to clinical ground only. Mice models are always difficult to be translated into human sarcomas

Mixofibrosarcoma. Authors again span from CDK6 amplification (that is found in less than 25% of the cases) to RAS dysregulation. I suggest authors to be more focused because the paper is not on cell cycle alterations but on CDK. Otherwise readers do not appreciate the true role of CDK in mixofibrosarcoma pathogenesis (at most marginal) and might expect CDK inhibitor activity where none has previously been observed.

Thank you for your thoughtful review and comments. While our review is ultimately focused on CDKs, we state at the outset in the Abstract that the review will consider alterations not only in the CDKs themselves but also the major CDK relevant pathways. Neither can be discussed at the exclusion of the other since CDKs and the cell cycle (and key regulators of the cell cycle, like Ras) are intrinsically linked. We believe that strictly focusing on CDK alterations would limit the insights to be gained from this review by basic researchers and clinical scientists interested in sarcoma biology and therapies. We include mouse models of sarcoma throughout the text as they are valuable (albeit imperfect) pre-clinical systems that complement clinical specimen analyses and enable the direct testing of molecular, biological and therapeutic hypotheses generated from patient tumor studies.

The reviewer is correct that Ras dysregulation (via genetic mutations) is not common in sarcomagenesis; however, the hyperactivation of Ras and its effector pathways (MAPK, PI3K/AKT) are frequent events driving these cancers. We apologize that this point was not made clearly enough in the original review and appreciate the opportunity to do so now. Thus, to address the reviewer’s concern and better clarify 1) the type of Ras dysregulation and 2) the essential functional link between excessive Ras signaling and CDK hyperactivation, we added more information into both the UPS and MFS sections (see the revised text below). The information provided in these and other early sections lays the conceptual framework for our subsequent discussion of CDK-targeted therapies, one of which is for MFS (see Table 3).   

Revised 2nd paragraph of UPS section:

Activation of Ras signaling, which results in excessive MAPK activity and consequent increased transcription of cyclin D1 [43, 44] and CDK4/6 [45, 46], has also been linked to UPS development [47, 48]. In a study containing 37 UPS patients, >80% displayed activated Ras/MAPK and PI3K/mTOR pathways. Notably, all but one of the tumors that were analyzed for mutations expressed wildtype NRAS, BRAF, and PIK3CA. Thus, while oncogenic mutations in these pathway genes are uncommon, pathway hyperactivation is frequent and predicts poor recurrence free and overall survival [47]. This prompted development of a conditional KrasG12D/+;p53flox/flox genetically engineered mouse model (GEMM). These double mutant mice develop UPS with 100% penetrance and accurately mimic the gene expression signature and lung metastatic features of the human disease. As p53-mediated tumor suppression is linked to upregulation of ARF (the alternative product of the Cdkn2a gene locus) by hyperactivated oncogenes like Ras [50-52], and mice lacking ARF primarily develop undifferentiated sarcomas [53], it was not surprising when Kirsch et al. demonstrated that conditional KrasG12D/+;Cdkn2aflox/flox mice efficiently developed UPS [54]. Other studies showed that intramuscular targeting of the Neurofibromatosis Type 1 (Nf1) gene, which encodes a negative regulator of Ras, combined with deletion of Cdkn2a leads to UPS formation [55]. These studies highlight the biological importance of excessive Ras signaling associated with CDK dysregulation in UPS biology.

4.2. Myxofibrosarcoma (MFS)

Until recently, myxofibrosarcoma was grouped among UPS but changes in morphologic and immunohistochemical criteria prompted their separation into distinct categories. Roughly 5% of sarcoma cases are classified as MFS, which is characterized by myxoid histology with hypocellular appearance and complex karyotypes [56]. MFS lesions display highly complex karyotypes that were recently deemed genetically indistinguishable from UPS tumors [57]. Although the two subtypes are genetically similar, MFS with at least 10% myxoid area generally display better prognosis than UPS. MFS lesions with less than 10% myxoid area display prognosis similar to that of UPS [58]. Further, MFS and UPS display differential locations. MFS are typically superficial and located in subcutaneous tissue (64.9%) whereas UPS lesions are almost always deep-seated below the muscle fascia (92.3%). Recent whole-exome sequencing of 99 MFS tumors revealed frequent alterations in genes related to the p53 and RB1 tumor suppressor pathways, including TP53, MDM2, RB1, CDKN2A/B, CCND1 (cyclin D1), and CDK6 [59]. CDK6 overexpression is observed in roughly 25% of MFS and is primarily driven by gene amplification on chr 7q. Increased CDK6 is associated with poor patient outcomes [60]. CDK6 shares 70% amino acid identity with CDK4. As such, the two proteins are for the most part considered functionally interchangeable although they often display differential, tissue specific expression [61, 62]. Additional mutations that have been observed in MFS include those causing inactivation of the NF1 gene, which causes heightened Ras signaling and upregulated transcription of the CDK6 partner, CCND1 [53, 64]. Increased cyclin D1 has been shown to drive tumorigenesis in NF1 mutant / Ras activated cancers [43, 44]. 

Reviewer 2 Report

This is a beautifully written review by Kohlmeyer et al. It describes in detail CDK function and de-regulation in soft tissue sarcomas. It also explores small molecule targeting of the pathway, on going clinical trials, and pre-clinical data. 

There are two small points to be addressed. In the UPS section on page 5 there should be mention of the KrasG12D/+; Ink4a/Arf fl/fl model developed by Kirsch et al, Nature Medicine 2007. This is a less well studied model but is relevant here. 

In the MFS section on page 5, the authors allude to the similarities/overlap between UPS and MFS. The authors neglect to mention or reference the finding from TCGA, Cell, 2017 "Comprehensive and Integrated Genomic Characterization of Adult Soft Tissue Sarcomas", wherein MFS is found to be genetically indistinguishable from UPS. 

Author Response

We are grateful to both reviewers for their time and effort considering our manuscript. We believe the changes made in response to their insightful comments have strengthened and clarified the review, which we hope will be beneficial to the sarcoma and CDK signaling fields.

Reviewer 2

This is a beautifully written review by Kohlmeyer et al. It describes in detail CDK function and de-regulation in soft tissue sarcomas. It also explores small molecule targeting of the pathway, ongoing clinical trials, and pre-clinical data. 

There are two small points to be addressed. In the UPS section on page 5 there should be mention of the KrasG12D/+; Ink4a/Arf fl/fl model developed by Kirsch et al, Nature Medicine 2007. This is a less well studied model but is relevant here. 

In the MFS section on page 5, the authors allude to the similarities/overlap between UPS and MFS. The authors neglect to mention or reference the finding from TCGA, Cell, 2017 "Comprehensive and Integrated Genomic Characterization of Adult Soft Tissue Sarcomas", wherein MFS is found to be genetically indistinguishable from UPS. 

Thank you for your thoughtful review and supportive comments.  In response to your first point, we added more discussion into the UPS section and properly referenced the Kirsch et al 2007 work describing the KrasG12D;Cdkn2aflox/flox mouse model.  We also agree with your second point and amended the text accordingly.  Specifically, the MFS section has been updated to reflect the findings from TCGA as well as a recent paper (Yoshimoto et al 2020 Am J Surg Pathol), which highlights the important clinicopathological differences between MFS and UPS.

Round 2

Reviewer 1 Report

Authors did not address the concerns 

Author Response

Thank you for your valuable comments.